# Lineage 7 Porcine Reproductive and Respiratory Syndrome Vaccine Demonstrates Cross-Protection Against Lineage 1 and Lineage 3 Strains

**DOI:** 10.3390/vaccines13020102

**Published:** 2025-01-21

**Authors:** Hsien-Jen Chiu, Shu-Wei Chang, Hongyao Lin, Yi-Chun Chuang, Kun-Lin Kuo, Chia-Hung Lin, Ming-Tang Chiou, Chao-Nan Lin

**Affiliations:** 1Department of Veterinary Medicine, College of Veterinary Medicine, National Pingtung University of Science and Technology, Pingtung 91201, Taiwan; b10516028@gmail.com (H.-J.C.); cychun0721@gmail.com (Y.-C.C.); kl.kuovet@gmail.com (K.-L.K.); 955302@gmail.com (C.-H.L.); 2Intervet Animal Health Taiwan Ltd., Taipei 11047, Taiwan; shu-wei.albert.chang@merck.com; 3MSD Animal Health Innovation Pte Ltd., Singapore 718847, Singapore; hongyao.lin@msd.com; 4Animal Disease Diagnostic Center, College of Veterinary Medicine, National Pingtung University of Science and Technology, Pingtung 91201, Taiwan; 5Research and Technical Center for Sustainable and Intelligent Swine Production, National Pingtung University of Science and Technology, Pingtung 91201, Taiwan

**Keywords:** PRRSV-2, modified live vaccines, heterologous challenge, cross-protection

## Abstract

**Background/Objectives**: Porcine reproductive and respiratory syndrome virus (PRRSV) has a major impact on swine productivity. Modified-live vaccines (MLVs) are used to aid in control. We investigated the cross-protection provided by a lineage 7 PRRSV MLV against a lineage 1 isolate under laboratory conditions and a lineage 3 challenge under field conditions in Taiwan. **Methods**: In the first study, thirty PRRS antibody-negative conventional piglets were vaccinated via the intramuscular (IM) or the intradermal (ID) route, with the control group receiving a placebo. Four weeks after immunization, all groups were challenged with a Taiwanese lineage 1 strain. The standard protocol for detection of reversion to virulence was applied to the vaccine strain in the second study, using sixteen specific pathogen-free piglets. In the third study, on an infected pig farm in Taiwan (lineage 3 strain), three hundred piglets were randomly selected and divided into three groups, each injected with either the PrimePac^®^ PRRS vaccine via the IM or the ID route, or a placebo. **Results**: In the first study, both vaccinated groups demonstrated reduced viraemia compared to the control group. The second study demonstrated that the MLV strain was stable. In the third study, piglet mortality, average daily weight gain, and pig stunting rate were significantly improved in the vaccinated groups compared to the control group. **Conclusions**: PrimePac^®^ PRRS is safe to use in the field in the face of a heterologous challenge, successfully providing cross-protection against contemporary lineage 1 and lineage 3 PRRSV strains from Taiwan.

## 1. Introduction

Porcine reproductive and respiratory syndrome virus (PRRSV) is an enveloped, positive-sense, single-stranded RNA virus of the family *Arteriviridae* [1]. It is the etiological agent of porcine reproductive and respiratory syndrome (PRRS) and causes major economic losses to the swine industry worldwide due to reduced growth rates, elevated mortality, and increased secondary bacterial infections [2]. One study estimated the costs of PRRS to the swine industry in the United States at USD 663 million per year in 2013 [3]. More recent studies have estimated this at USD 1.2 billion [4]. PRRS viruses are divided into two distinct species, Betaarterivirus suid 1 (PRRSV-1) and Betaarterivirus suid 2 (PRRSV-2) [1], with the former originating from the former Eastern Europe and the latter originating from North America [5]. PRRS viruses have a high level of genetic diversity, and consequently, PRRSV-1 and PRRSV-2 are further categorized into four subtypes [5] and 11 lineages [6], respectively, according to phylogenetic analysis of the ORF5 region. PRRSV-2 has a high level of genetic diversity, and lineage 1 has become the most prevalent and diverse lineage in Asia and the United States. In China, lineage 1 strains have become dominant [7], with NADC30-like strains being prominent [8,9]. In Thailand, South Korea [10], and Taiwan [11], lineage 1 strains have also been reported; however, lineage 3 isolates are predominant in Taiwan [12]. PRRSV-1 isolates have also been reported to circulate in Asia [10,13], and it is likely that farms may be simultaneously infected with both virus species.

While PRRSV control at the local level relies on biosecurity [14], in high-pig-density regions, where the infection is enzootic, with frequent emergence of variant strains, the main tool available to producers to mitigate the economic losses induced by PRRSV is vaccination. In Asia, numerous PRRS modified-live vaccines (MLVs) are commercially available, which can be based on VR2332, R98, NEB-1, P129, CH-1R, HuN4-F112, or JXA1-P80 strains. Vaccine strains VR2332, CH-1R, and R98 are of lineage 5 [15]; NEB-1 is of lineage 7 [16]; and the others belong to lineage 8 [17]. Several authors have advocated for MLV selection criteria based on an “as close as possible” approach, relying on genetic or antigenic matching of field strains with the vaccine strain to be chosen. In practical terms, this means selecting an MLV based on a strain belonging to the same lineage as that detected on the farm. However, infection at farm level with multiple strains makes this approach more and more difficult to implement, more so when co-infection with PRRSV-1 and PRRSV-2 strains is detected [18,19]. Generally, inactivated PRRS vaccines provide poor protection against both homologous and heterologous strains [20], while modified-live vaccines can provide adequate protection in a homologous challenge.

The literature provides limited and sometimes conflicting information on cross-protection from existing MLV vaccines against heterologous NADC30 or NADC34 isolates. For instance, reports have shown that a VR2332-based MLV is able to induce cross-protection against a lineage 1 strain upon challenge [21], while others have shown that VR2332-based vaccines provide limited cross-protection against lineage 1 strains [22]. Clear information on cross-protection is necessary for veterinarians to decide on PRRS control measures in the field. A new commercial lineage 7 MLV (PrimePac^®^ PRRS, MSD Animal Health, Rahway, NJ, USA) has recently become available in Taiwan. Both lineage 1 and lineage 3 are prevalent in Taiwan. To understand the cross-protection against lineage 1 strains, the present study was performed under both laboratory and field conditions in order to understand whether a commercial lineage 7 MLV would be able, through cross-protection, to reduce physiological and economic losses associated with field lineage 1 and lineage 3 strains from Taiwan.

## 2. Materials and Methods

The animal experimental procedure was reviewed and approved by the Institutional Animal Care and Use Committee (IACUC) of the National Pingtung University of Science and Technology (NPUST) with approval number NPUST-111-070. The vaccine used was a commercially available lineage 7 MLV (PrimePac^®^ PRRS, MSD Animal Health, Rahway, NJ, USA, Batch Number A606CE05). It was prepared according to the instructions of the manufacturer and injected either via intradermal (ID) injection (each dose reconstituted to 0.2 mL with Diluvac Forte^®^ (MSD Animal Health, Rahway, NJ, USA), minimum 4.0 log10 TCID_50_ per dose) using a purpose-built intradermal injector supplied by the vaccine manufacturer (IDAL^®^, MSD Animal Health, Rahway, NJ, USA) or via intramuscular (IM) injection (each dose reconstituted to 1 mL with Diluvac Forte^®^ (MSD Animal Health, Rahway, NJ, USA), minimum 4.0 log10 TCID_50_ per dose).

### 2.1. In Study 1 (Laboratory Efficacy Trial)

Thirty 3-week-old PRRS antibody-negative conventional (high health) piglets were randomly allocated to one of the three following groups (10 piglets per group): i. intramuscular (IM) injection of a dose (1 mL) of a commercial lineage 7 MLV (PrimePac^®^ PRRS, MSD Animal Health, Rahway, NJ, USA) into the right side of the neck, ii. intradermal (ID) injection into the right side of the neck of a dose of the same vaccine (0.2 mL), or iii. IM injection of the vaccine diluent (1 mL) into the right side of the neck muscle. Vaccinated piglets were kept in the same facility; control (diluent-injected) piglets were housed in a separate facility. At 7 weeks of age (4 weeks after immunization), all groups were challenged with 2 mL of a 1 × 10^4^ TCID_50_/mL suspension of a Taiwanese PRRSV strain (lineage 1 NADC34-like strain, GenBank accession no. OM779010), with 1 mL via the nasal route and 1 mL via the IM route. All pigs were monitored daily for clinical signs. Control (challenged) pigs remained in a different facility from the other two groups. Two weeks after challenge (6 weeks post immunization), all pigs were humanely euthanized and necropsied (Figure 1A).

All pigs were individually weighed before immunization, before inoculation and before euthanasia, and the average daily weight gains were compared between groups (Figure 1A). The rectal temperature of all pigs was measured before challenge, and daily after challenge. Mortality events and respiratory clinical symptoms were recorded on a daily basis. Respiratory clinical scoring was performed in compliance with the scoring previously described for PRRSV-1 and PRRSV-2 pathogenicity comparison [23]. Briefly, individual pigs were scored from 0 (no apparent clinical signs) to 6 (severe dyspnea/tachypnea at rest). Gross lung lesions were assessed in all pigs, using the method detailed in the same publication. The calculation used for lung lesion percentage is 100 × {(0.10 × left apical lobe) + (0.10 × left cardiac lobe) + (0.25 × left diaphragmatic lobe) + (0.10 × right apical lobe) + (0.10 × right cardiac lobe) lobe) + (0.25 × right diaphragmatic lobe) + (0.10 × middle lobe)}. Individual blood samples were taken from all pigs on the day of immunization, 2 weeks later, before challenge (4 weeks after immunization), and 3 days, 7 days, 10 days, and 14 days after the challenge. Sera were collected and frozen (−80 °C) until they were processed. Once thawed, reverse-transcription quantitative PCR (RT-qPCR) was performed on the sampled sera, in compliance with the protocol detailed elsewhere [24]. Serum anti-PRRSV antibodies were detected using commercially available ELISA kits (PRRS X3^®^ Ab Test, IDEXX, ME, USA). These kits allow for the determination of antibody titers. All sera were also processed by indirect immunofluorescence assay (IFA). Briefly, MARC-145 cells were cultured in 96-well culture plates. First, they were inoculated with 100 TCID_50_/well of a PRRSV suspension. Sera were serially diluted with a phosphate buffer solution (PBS) and then added to the cell plate wells. After induction for 1 h, wells were rinsed three times with PBS, then the FITC-conjugated goat anti-swine IgG antiserum was added. After 1 h incubation, the wells were rinsed three times with PBS, and residual fluorescence was measured under UV-light microscopy. Lung tissue and hilar lymph nodes were collected from all necropsied pigs. After homogenizing, genomic load was measured by RT-qPCR, also in compliance with a previously described method [24].

### 2.2. In Study 2 (Laboratory Reversion to Virulence Trial)

Sixteen specific pathogen-free (SPF) piglets aged 3 to 5 weeks old, negative for PRRSV genome and antibodies, were used in pairs. The first pair was intranasally inoculated with 1 mL of 10 times the registered vaccine dose of the PRRS Nebraska MLV strain (10^5^ TCID_50_/mL or more). The two piglets were sacrificed 14 days later, and 50 mL PBS were injected into the trachea; after flushing, the bronchoalveolar lavage fluid was collected (about 30 mL were recovered) and an aliquot was tested by RT-qPCR. Positive specimens were concentrated (centrifuged at 2000× *g* for 10 min at 4 °C, after which the supernatant was removed and a bronchoalveolar lavage pellet was obtained) and stored at −80 °C until further use. The pellet was thawed and dissolved in 3 mL PBS. PRRSV titer was determined by RT-qPCR, and the suspension was aliquoted for further passage in another SPF pair of piglets (four pairs in total). The fourth-passage pellet was similarly resuspended and inoculated into 8 SPF piglets, which were sacrificed 21 days later. Again, PRRSV was harvested by bronchoalveolar lavage. All piglets were monitored daily for clinical signs and mortality, and lung scoring was performed at necropsy. Bronchoalveolar lavage fluids were used for virus isolation and viral genome detection (Figure 1B).

### 2.3. In Study 3 (Field Efficacy Trial)

A PRRS-positive pig farm in Taiwan with a lineage 3 isolate was selected to conduct the field trial. Identification of the different PRRSV lineages was based on the complete open reading frame 5 sequences [6]. The farm selected was a single-site 1200-sow farrow-to-finish production farm located in the southern area of Taiwan. No PRRS vaccination was used on this farm at the time of the study. Three hundred 2-week-old piglets were randomly selected and divided into three groups of 100 piglets each and individually marked (ear tags) with different colors (one color per group). In the immunized groups, pigs were injected with the PrimePac^®^ PRRS vaccine (MSD Animal Health, Rahway, NJ, USA), (lineage 7) into the right neck muscle (1 mL) or intradermally (0.2 mL), with 100 pigs per group. In the control group, 100 pigs were sham-vaccinated (vaccine diluent, 1 mL via the IM route). All 300 piglets were comingled to avoid introducing a bias that might be associated with farm management; stockmen were blind to the allocation of the color of earmarks to the treatment group. Pigs were monitored daily for clinical symptoms and mortality until 12 weeks of age. Necropsy was performed on all dead animals. Blood samples were collected from 30 piglets per group before immunization (2 weeks of age) and then 2, 4, 6, 8, and 10 weeks later (Figure 1C). The same RT-qPCR and ELISA techniques were used as mentioned in the efficacy trial section. All pigs were individually weighed at 2, 4, and 12 weeks of age (Figure 1C). Mortality rate, average daily gain (ADG), serum viremia, and ELISA titers were measured. In addition, any dead pigs were necropsied and gross lesions as well as tissue PRRSV loads were assessed.

### 2.4. Statistical Analysis

Normal distributed and continuous variables such as body weight, average daily weight gain, viremia, serum antibody titer, and virus loads in the lungs and hilar lymph nodes were analyzed by variance analysis (ANOVA). With the macroscopic lung lesion score and the lung tissue lesion score being continuous variables, scoring results were compared between multiple groups by the Kruskal–Wallis test. Tukey’s test was used to conduct post hoc tests between the two groups in both the field trial and the laboratory efficacy study. Discrete variables (mortality rate, virus positivity rate, clinical respiratory symptom score, lung gross lesion score, and pig dysplasia rate) were analyzed by Chi-square test (pig stunting rate is defined as the proportion of pigs whose weight is below 75% of the average pig weight of the group). Where the number of samples was small (reversion to virulence test), Fisher’s exact test was used.

**Figure 1 vaccines-13-00102-f001:**
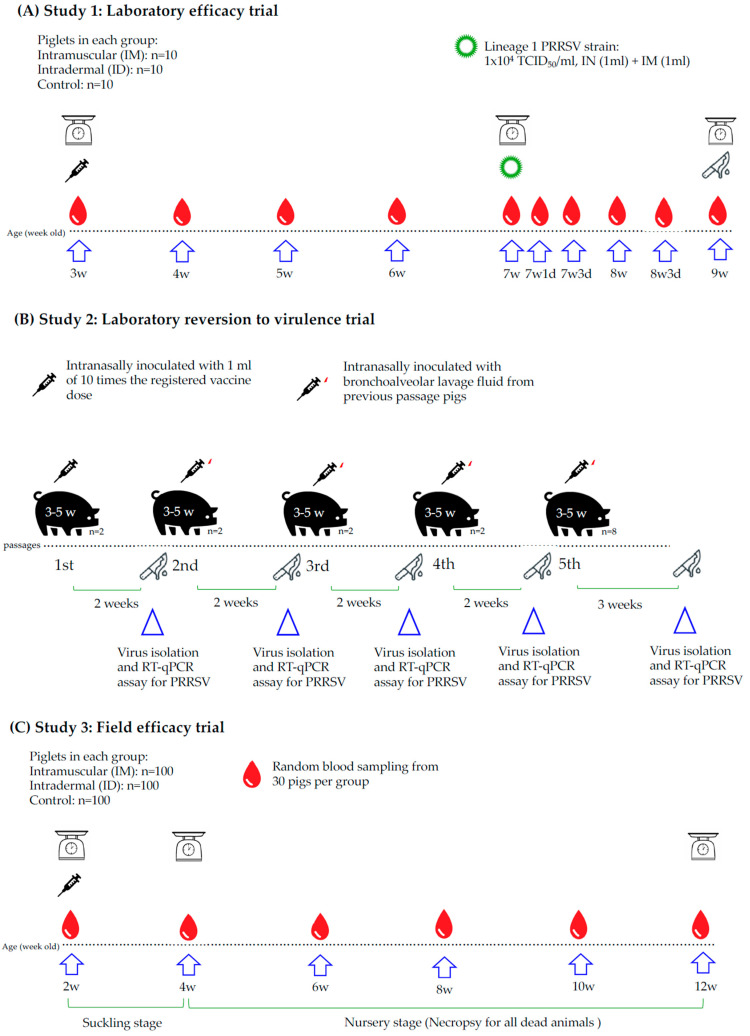
Scheme of the study design. (**A**) Laboratory efficacy trial; (**B**) laboratory reversion to virulence trial; (**C**) field efficacy trial.

## 3. Results

### 3.1. Laboratory Efficacy Trial

#### 3.1.1. Clinical Observations

All individual rectal temperature values after vaccination remained below the threshold defining hyperthermia (40.5 °C, Figure 2). On the first day post challenge (DPC1), fever (40.9 °C) was observed in one pig in the control group. In the IM group, fever (40.7 °C) was observed in a single pig on the second day after challenge (DPC2). The average body temperature of the IM group was indicative of hyperthermia only on DPC6. In the ID group, fever was observed in one pig (40.7 °C) on DPC3, followed by sporadic fever, and the average body temperature in the ID group was never higher than 40.5 °C (Figure 3).

#### 3.1.2. Respiratory Signs

Only mild respiratory clinical symptoms were observed post challenge, on DPC9 and DPC12, as detailed in Table 1. No respiratory clinical signs were observed in the ID group. Respiratory signs were observed in the control group from DPC6 to the end of the trial, and the highest score was 2.

#### 3.1.3. Comparison of Body Weight and Average Daily Gains

The body weights and average daily gains of pigs at different stages are presented in Table 2. No significant differences were evidenced between groups.

#### 3.1.4. Lung Lesion Scores

There was no significant difference in gross or microscopic lung lesion scores between groups (*p* > 0.05) (Figure 4A). There was significant difference in microscopic lung lesion scores between ID and control groups (*p* < 0.05) (Figure 4B).

#### 3.1.5. Viremia

Vaccination induced a limited viremia in both IM and ID groups (absent in the control group), which remained at a constant level until challenge. Viremia associated with the challenge strain began to be detected on DPC3 in the vaccinated groups, while it was detected as early as DPC1 in the control group (Figure 5). Both the IM and ID groups had similar levels of viremia, without significant difference at any point post challenge. In all three groups, viremia peaked on DPC7 and then gradually subsided. The average serum genomic load of the IM group was significantly lower than that of the control group on DPC1, DPC7, DPC10, and DPC14 (*p* < 0.05). The average serum genomic load of the ID group was consistently significantly lower than that of the control group (*p* < 0.05).

#### 3.1.6. Serum Antibodies

Pigs in the vaccinated groups began to produce detectable anti-PRRSV antibodies 14 days after immunization (DPV14), while antibodies were detected by ELISA in the control group from DPC7 onwards. The average antibody titers in the vaccinated groups were significantly higher than those in the control group at DPC1, DPC3, and DPC7 (*p* < 0.05). In addition, the average antibody titers in the ID group were significantly higher than those in the IM group at DPV 14, DPV21, DPC1, and DPC3 (*p* < 0.05) (Figure 6). Furthermore, the increase in antibody titers observed in the vaccinated groups from DPC7 onwards coincided with peak viremia and is suggestive of an anamnestic response in these pigs. No piglet was found to be positive for IFA antibodies on the day of vaccination. Four weeks after vaccination (DPV28), the average IFA titer of the IM group was 5.3 ± 1.06 log2, and that of the ID group was 5.5 ± 1.35 log2. Although the average titer of the ID group was slightly higher than that of the IM group, they were not significantly different (*p* > 0.05), while the control group was still negative on DPV28.

#### 3.1.7. Tissue Viral Loads

The viral load of the tested tissues (homogenates of lungs and hilar lymph nodes) is expressed as a gene ratio (number of PRRSV ORF5 divided by the genomic load of a pig housekeeping gene). The average ratio of the ID group was significantly lower than that of the IM group (*p* < 0.0005). The average ratio of the IM group was slightly higher than that of the control group, but there were no significant differences between the latter ones (*p* > 0.05) (Figure 7).

### 3.2. Laboratory Reversion to Virulence Trial

The bronchoalveolar lavage fluid collected from the pairs of pigs at the first and second passages was negative in both virus isolation and PRRSV nucleic acid detection. Blood collected and tested via qPCR from the first passage was positive at CQ level 38 in one animal, with the rest negative. In the second passage, blood samples were all qPCR negative. In the interest of animal welfare outcomes and to reduce the number of trial animals sacrificed, the third to fifth passage tests were not performed.

### 3.3. Field Efficacy Trial

#### 3.3.1. Mortality Rate

There were no deaths or adverse reactions in any group of field pigs between 2 weeks of age (vaccination) and 4 weeks of age (weaning). The peak mortality occurred in the middle nursery period (7–8 weeks of age) in all groups, with 10 pigs in each of the IM and ID groups and 16 pigs in the control group (Figure 8). Over the nursery stage, the mortality rate in the IM group was 13% (13/100), while it was 12% in the ID group and 24% in the control group (Figure 8). There was no statistically significant difference in mortality rates between the IM and the ID groups (*p* > 0.05). The mortality rate of the control group was significantly higher than that of both vaccinated groups (*p* < 0.05).

#### 3.3.2. Average Daily Weight Gain

There was no significant difference in the average body weight of each group at 2 weeks of age (vaccination) and 4 weeks of age (weaning) (*p* > 0.05). However, at 12 weeks of age (when the pigs leave the nursery), the average body weight of the pigs in the IM and ID groups was significantly higher than that of the control group (*p* < 0.05) (Figure 9). The average daily weight gain was not significantly different between groups (*p* > 0.05) while in the farrowing unit (2–4 weeks old). However, over the nursery stage (4–12 weeks old), the average daily weight gain of the IM and ID groups was significantly higher than that of the control group (*p* < 0.05). The average body weight of all field pigs at 12 weeks of age was 24.91 kg, and the number of pigs below 75% of that value (18.68 kg), i.e., stunted pigs, was 8% (7/87) in the IM group, 5.6% (5/88) in the ID group, and 18.4% (14/76) in the control group. The proportion of stunted piglets was significantly lower in the vaccinated groups than in the control group (*p* < 0.05), without a statistically significant difference between the IM and the ID groups (*p* > 0.05).

#### 3.3.3. PRRSV Viremia

All pigs were RT-qPCR negative on the day of vaccination. At weaning (4 weeks of age), viremia was detected in all three groups, indicating either that a wild-type PRRSV strain was circulating in the farrowing unit, or that the MLV strain was being shed by the vaccinated piglets, or both. Viremia levels and the proportion of positive pigs peaked in the middle stage of the nursery (6–8 weeks of age) regardless of group. In the IM and control groups, the peak was reached at 6 weeks of age. In one control pig, viremia reached 6.63 log10 copies/μL at 6 weeks of age, which was the highest load detected in this field trial. The ID group peaked at 8 weeks of age (3.93 ± 0.89 log10 copies/μL). Regarding the proportion of positive (viremic) pigs within groups, both the ID and the control groups reached their maximum at 8 weeks of age (86.7 and 100%, respectively). The IM group reached its maximum positivity rate at 10 weeks of age (93.3% of positives). There was no statistically significant difference in viremia and positivity rates among the three groups at any stage (*p* > 0.05) (Figure 10).

#### 3.3.4. Anti-PRRSV Antibodies

During the field trial, the average antibody titer profiles were similar between groups. The average antibody S/P values at 2 weeks of age were 1.5 (Figure 11). These values dropped to their lowest point at 5 weeks of age (S/P value of about 0.9), suggesting a decay in maternally derived immunity. Then, they began to rise and peaked in the later stages (10–12 weeks of age, S/P value of about 2.2) (Figure 11).

#### 3.3.5. Gross Lesions and Tissue Genomic Loads of Necropsied Pigs

Thirteen pigs died in the IM group, 12 in the ID group, and 24 in the control group (Figure 8). The gross lesions of the dead pigs were mainly polyserositis caused by bacterial co-infections or Mycoplasma infection, followed by viral pneumonia. Molecular biology and bacteriology performed from sampled lesions detected pathogens such as *Mycoplasma hyorhinis* (*M. hyorhinis*), *Glaesserella parasuis* (*G. parasuis*), *Streptococcus suis* (*S. suis*), *Streptococcus* sp., and *Pasteurella multocida* (*P. multocida*). The PRRSV genomic load in the lung tissue of dead pigs was highest in the control group (6.55 ± 0.91 log10 copies/μL), followed by the ID group (6.43 ± 0.63 log10 copies/μL) and the IM group (6.09 ± 1.28 log10 copies/μL). These values were not significantly different (*p* > 0.05).

## 4. Discussion

The on-farm PRRS disease situation in Asia is complex, often with multiple isolates present on farms [18,19]. Vaccination is a commonly used strategy to mitigate the effects of PRRS infection on farms. Regulatory approval of new PRRS vaccines is a complex process that assesses safety and efficacy, often taking years before granting new vaccines commercial licenses [25]. Given the high mutation rate of the PRRS virus resulting in high variability and diversity, the availability of new commercial PRRS vaccines struggles to keep up with the strains of the day. It is therefore difficult for field veterinarians to find homologous commercially available MLV strains—in particular, against the NADC30 or NADC34-like strains. Ideally, a PRRS MLV that exhibits cross-protection against heterologous strains will demonstrate efficacy by decreasing clinical symptoms such as viremia and lung lesions and hence improve production parameters such as average daily gain or mortality against a control group that has not received the vaccine [3,4]. In the field, a common approach by veterinarians is to use existing commercial MLVs with matching genotype but different lineages to achieve cross-protection. It is hence necessary to investigate whether commercial modified-live vaccines can provide cross-protection against contemporary circulating PRRS strains of different lineages. When designing such studies, vaccines should be evaluated for both safety and efficacy under laboratory and field conditions. Safety is key when using PRRS modified-live vaccines, as recombination events have been detected between field and vaccine strains [26,27] and even between MLV strains [28]. In terms of efficacy, vaccines may perform differently under laboratory and field conditions. The higher density of pigs in the field contributes to elevated disease pressure [29], as well as to potential co-infection with other pathogens, such as *G. parasuis* or *S. suis*, a situation that is difficult to replicate under laboratory conditions [30]. It is therefore important to evaluate vaccines in both the field and the laboratory to understand their efficacy. We therefore designed the present study to evaluate the safety and protection provided by this lineage 7 MLV under laboratory conditions against a lineage 1 strain and under field conditions against a lineage 3 strain.

From the vaccine safety perspective, the present trial confirmed that the PrimePac^®^ PRRS MLV strain has minimal shedding when administered to pigs compared to another MLV strain [31]. Under both laboratory and field conditions, pigs vaccinated with PrimePac^®^ PRRS did not exhibit hyperthermia or adverse reactions, also confirming the high safety level reported for NEB-1, including regarding limited secondary bacterial infection levels in vaccinated field piglets [16]. No significant differences were observed in any of the measured outcomes between pigs vaccinated by the IM or the ID routes. These characteristics (limited shedding and very limited adverse reactions) should provide assurance to veterinarians.

From the vaccine efficacy perspective, the laboratory trial showed that vaccination induces a rise in specific anti-PRRSV antibodies and an immune protection that limits both clinical signs and viremia upon a heterologous challenge. However, ADG and microscopic lung lesion scores were not significantly different between groups under the good sanitary conditions of the experimental controlled environment (Study 1). In the field trial (Study 3), with a “natural” challenge of the same vaccine by a lineage 3 strain, pig mortality rate, average daily weight gain, and pig stunting rate were significantly improved in the vaccinated (IM and ID) groups compared to the control group (vaccine diluent, IM). We believe that the difference in the latter outcomes, as compared to the laboratory trial, is likely due to the higher swine density and greater infectious pressure of secondary bacterial infections [29]. This is further supported by our necropsy results on pigs that died of clinical respiratory disease in the field study. We found that most pigs necropsied were found to be co-infected with PRRS and secondary bacterial infections caused by *M. hyorhinis*, *G. parasuis*, *S. suis*, *Streptococcus* sp., and/or *P. multocida*. Work by other authors supports that PRRSV infection alone may be mild, but clinical signs and impact on productivity can be greatly potentiated by secondary bacterial infections [2,32,33,34], coming together into porcine respiratory disease complex (PRDC). These observations also support the finding that a major impact of PRRS is the increased use of antibiotics to control secondary bacterial infection [35].

In terms of protection in the face of maternally derived antibodies (MDAs), our field study was controlled to ensure that all animals in the study had similar initial PRRS ELISA titers of an S/P ratio of 1.5 at the 2-week mark when they were vaccinated. Other studies have found that at levels between SP 1.5 and 2, the development of the antibody response was delayed or reduced [36]. Despite this potential interference by MDAs [37], the pig mortality rate, average daily weight gain, and pig stunting rate in our study were significantly improved in the vaccinated (IM and ID) groups. This is potentially due to the protection still afforded by the vaccine due to cell-mediated immunity [38,39], which is not impacted by MDAs. In the field, the nursery is the peak PRRS infection period, as piglets of different immune status are comingled. Hence, producers may still opt to vaccinate at the 2-weeks-of-age mark when piglets are still in the farrowing crates, MDA levels are high, and before the peak of infection in the nursery. Our study hence shows that there is still value in vaccinating in the face of medium to high maternal-derived antibodies in the field to reduce production losses due to PRRSV infection.

Pigs vaccinated by the ID route presented significantly lower average genomic PRRSV loads in the blood and hilar lymph nodes than those vaccinated via the IM route. They also developed consistently significantly higher ELISA antibody titers against PRRSV. The ID-vaccinated piglets also developed an earlier and more limited peak of infection upon heterologous challenge, in agreement with a similar observation by other authors [40]. The mechanism for this is supported by the work of multiple authors [31,41,42] where ID vaccination induced faster development of cell-mediated immunity compared to the IM route. An improved cell-mediated immune response has been demonstrated to reduce clinical symptoms in PRRSV-infected piglets and sows [37,38] through potentially faster clearance of viremia post infection. Our findings in the laboratory study where lung lesion scores could be measured found that ID-vaccinated pigs had significantly lower lung lesion scores compared to the control group, whereas the IM-vaccinated pigs did not reveal a significant difference, providing a clinical correlation and consequence of improved viremia control.

Despite these findings, in the laboratory and field studies, differences in immune outcomes between the IM and ID groups did not lead to overall differences in long-term production outcomes. This is explained by the lack of secondary bacterial infections potentiating PRRS infections [2,33,34,35] in the laboratory study, resulting in relatively similar outcomes between IM and ID outcomes. In the field study, it is possible that because pigs were co-mingled, differences between groups may not have been as evident. In a real field situation, all pigs would be vaccinated to a similar level and there would not be “seeder” or intentionally non-vaccinated animals present to artificially increase the infectious pressure on the farm. Future study designs will take this into account. Despite the lack of finding long-term production outcomes in our study, other studies have detailed additional benefits of ID vaccination, which include reduced stress and muscle damage—ultimately contributing to improved animal welfare [42] and reduced iatrogenic transmission of pathogens, and contributing to improved farm biosecurity [43]. The use of ID vaccination may thus provide other results for swine producers compared to the IM route.

In both the laboratory and the field trials, we acknowledge the limitation of the RT-qPCR of our laboratory not being able to distinguish between vaccine or field strains. Several authors have demonstrated a differential RT-qPCR method that would have been able to distinguish the viremia titer changes of PRRSV lineage 7 vaccine strain and PRRSV lineage 3 wild strain in the field study [44]. To the best of our knowledge, we were not able to identify authors who have carried out work to distinguish PRRSV lineage 7 vaccine strain or PRRSV lineage 1 strain in viremia. One possible confounding factor may hence be the possibility of vaccine virus causing disease in vaccinated animals. In our opinion, this is unlikely, as other authors [16,31] who have worked with this similar vaccine isolate have found that the viremia and shedding induced by the vaccine strain is low and short-lived compared to field strains. Our study findings also conclusively support this in Study 1 and Study 2. We found that vaccine virus administered alone exhibited a short shedding period, did not revert to virulence after passage, and did not cause clinical symptoms such as pyrexia or respiratory symptoms in the absence of viral challenge. Separately, in Study 3, given that only the vaccinated groups (ID and IM) demonstrated a reduction in mortality and stunting rate and that there was an increase in ADG despite viremia levels being statistically similar across all three groups, the lineage 7 vaccine virus strain provides clinical benefits and protection against lineage 3 isolates in the field without causing problems in itself.

Another limitation of our study was the lack of lung lesion evaluation in the three groups of pigs in the field study. In Taiwan, a majority of pigs are individually sold through a wholesale auctioning system in the slaughterhouse [45]. This caused difficulties in tracing the groups to the slaughterhouse. We attempted to compensate for this by carrying out the lung lesion scoring in the laboratory study instead. Finally, farm management did not allow for tracing of the total volume of antibiotics used for treatment in the groups of pigs, which prevented the assessment of the economic value of vaccination in this context. These aspects will be added to the inclusion criteria for farms in further trials.

## 5. Conclusions

In the complicated context of PRRSV infection in Asia, where multiple NADC30- or NADC34-like (lineage 1) strains are co-circulating with other lineage strains (and sometimes with PRRSV-1 strains), science-based information on cross-protection provided by commercially available MLV strains is highly desirable from a prescriber point of view. The present studies demonstrated that the NEB-1 (lineage 7) strain, used in the PrimePac^®^ PRRS MLV, is both stable and safe in pigs. The vaccine induces cross-protection in a heterologous challenge with field strains under both controlled (lineage 1) and field (lineage 3) conditions. In the latter trial, ID and IM routes of vaccination significantly improved production parameters such as mortality compared to the non-vaccinated group. ID-vaccinated pigs developed earlier infection control than the IM-vaccinated ones, but this did not result in a significant difference in performance between the two experimental groups.

## Figures and Tables

**Figure 2 vaccines-13-00102-f002:**
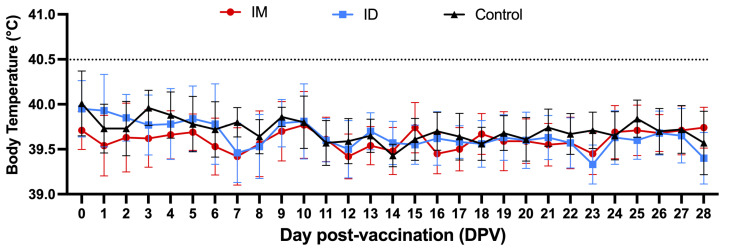
Daily average rectal temperature after vaccination per experimental group. The dotted line indicates the threshold for a clinical definition of fever (40.5 °C).

**Figure 3 vaccines-13-00102-f003:**
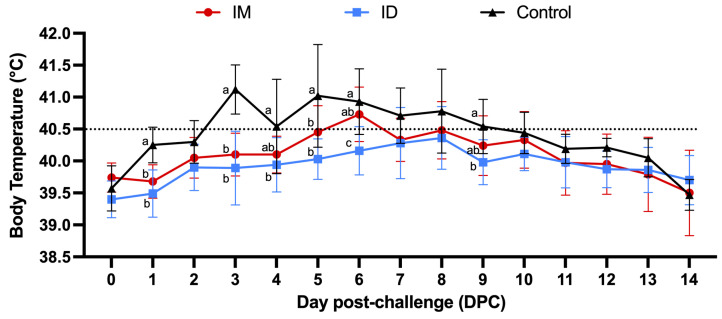
Daily average rectal temperature after the challenge. The dotted line indicates the threshold for a clinical definition of fever (40.5 °C). Different letters indicate significant differences between each group at a given date (*p* < 0.05).

**Figure 4 vaccines-13-00102-f004:**
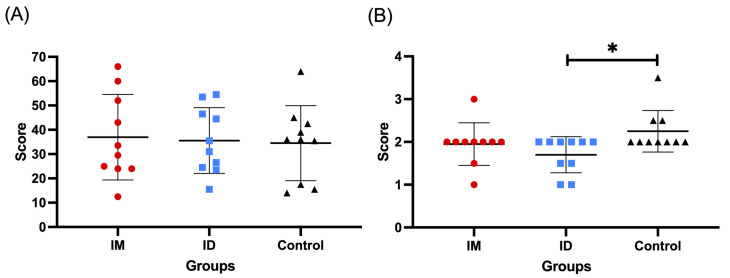
Macroscopic (**A**) and microscopic (**B**) lung lesion scores. * Indicates significant differences between groups (*p* < 0.05).

**Figure 5 vaccines-13-00102-f005:**
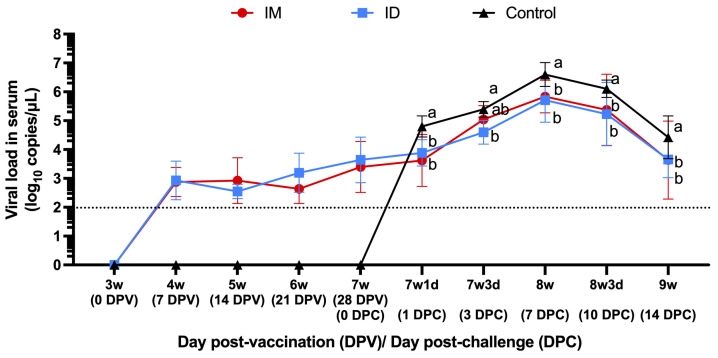
Genomic PRRSV load in the serum of pigs after vaccination (day post vaccination, DPV) and after challenge (day post challenge, DPC). Different letters indicate significant differences between each group (*p* < 0.05) on a given date. The dotted line is the detection limit of PRRSV RT-qPCR.

**Figure 6 vaccines-13-00102-f006:**
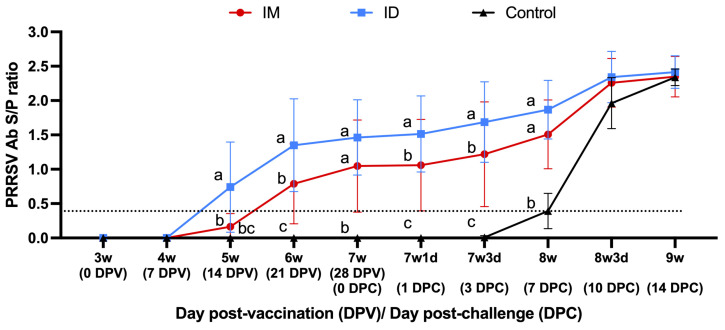
Elisa PRRSV-specific antibody values in the serum of pigs after vaccination (day post vaccination, DPV) and after challenge (day post challenge, DPC). Different letters indicate significant differences between each group at a given date (*p* < 0.05). The dotted line is the PRRSV ELISA interpretation threshold (0.4); any point below is considered negative.

**Figure 7 vaccines-13-00102-f007:**
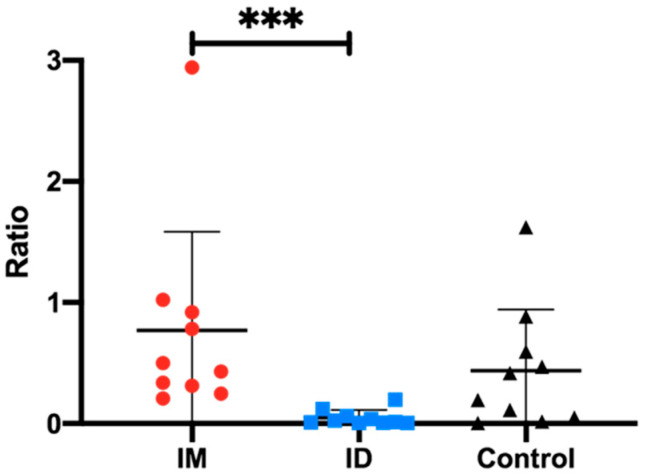
Comparison of the viral genomic load in lung tissue and hilar lymph nodes of pigs in each experimental group. Red: IM group; blue: ID group; black: control group. *** Indicates significant differences between groups (*p* < 0.05).

**Figure 8 vaccines-13-00102-f008:**
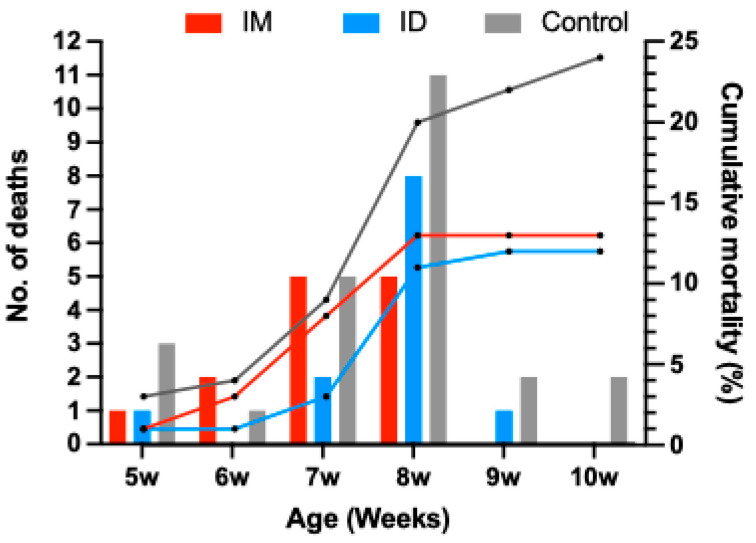
Comparison of the weekly number of dead pigs and cumulative mortality rate in each group during the nursery stage. The *y*-axis on the left is the number of dead animals, and the *y*-axis to the right is the cumulative mortality rate among each group.

**Figure 9 vaccines-13-00102-f009:**
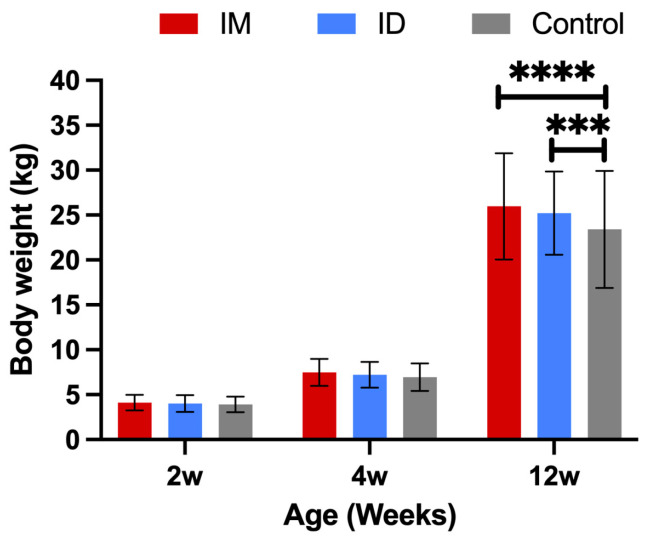
Comparison of the liveweight of pigs in each group at 2, 4, and 12 weeks of age. ****, *** Indicate significant differences between groups (*p* < 0.05).

**Figure 10 vaccines-13-00102-f010:**
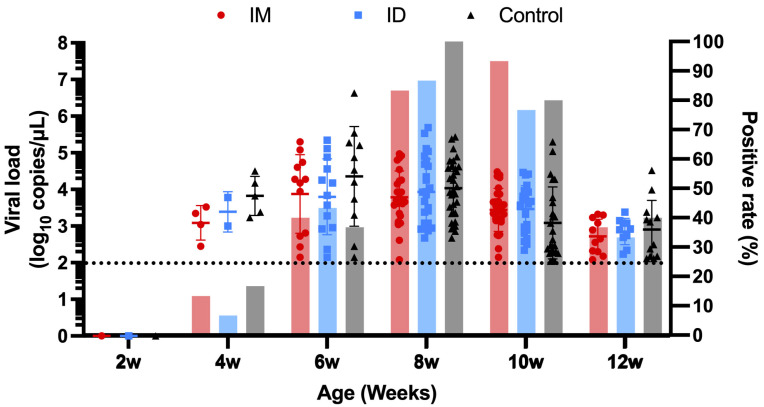
Comparison of the amount of PRRSV in the serum of pigs of different groups at different ages. The *y*-axis on the left is the genomic load, and the *y*-axis to the right is the virus positivity rate among each group. The dotted line is the limit of detection of the PRRSV RT-qPCR.

**Figure 11 vaccines-13-00102-f011:**
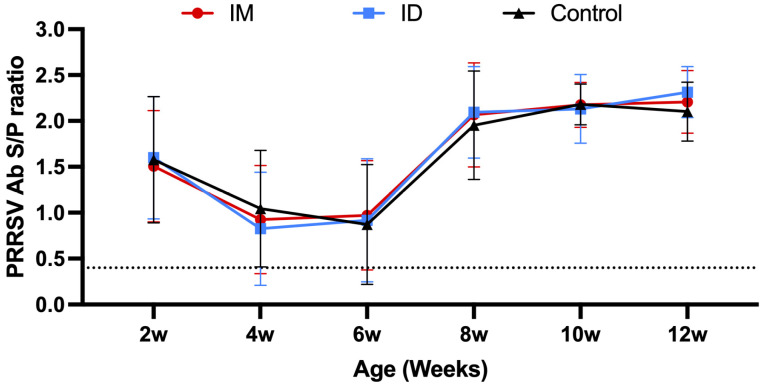
Elisa PRRSV-specific antibody values in the serum of field pigs. The dotted line is the PRRSV ELISA interpretation threshold (0.4); any point below is considered negative.

**Table 1 vaccines-13-00102-t001:** Respiratory clinical scores in each experimental group of the efficacy trial.

Group	DPC														
	0	1	2	3	4	5	6	7	8	9	10	11	12	13	14
IM	0	0	0	0	0	0	0	0 ^a^	0	0.2 (0–2)	0	0	0.2 (0–2)	0 ^a^	0 ^a^
ID	0	0	0	0	0	0	0	0 ^a^	0	0	0	0	0	0 ^a^	0 ^a^
Control	0	0	0	0	0	0	0.2 (0–2)	1 ^b^ (0–2)	0	0.4 (0–2)	0.2 (0–2)	0.4 (0–2)	0.2 (0–2)	0.8 ^b^ (0–2)	1 ^b^ (0–2)

The range of scores is indicated in parentheses. Different letters indicate significant differences between groups (*p* < 0.05) on a given date.

**Table 2 vaccines-13-00102-t002:** Liveweight and average daily weight gain measured for each experimental group of the efficacy trial at different ages/production periods.

Group	Body Weight (kg)			Average Daily Weight Gain (g/day)		
	3 d	7 W	9 W	3 W–7 W	7 W–9 W	3 W–9 W
IM	6.44 ± 0.77	13.23 ± 2.09	19.26 ± 2.44	242.50 ± 55.8	430.78 ± 97.89	305.26 ± 42.56
ID	6.28 ± 0.59	11.51 ± 1.58	17.56 ± 2.67	186.61 ± 56.44	432.78 ± 87.66	268.66 ± 60.92
Control	6.34 ± 0.57	11.58 ± 1.92	17.88 ± 2.31	187.39 ± 55.39	449.42 ± 47.42	274.73 ± 45.21

## Data Availability

The data presented in this study are available on request from the corresponding author.

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
