# Peer review of "Lineage 7 Porcine Reproductive and Respiratory Syndrome Vaccine Demonstrates Cross-Protection Against Lineage 1 and Lineage 3 Strains"

_vaccines, 2025, doi:10.3390/vaccines13020102_

Round 1
Reviewer 1 Report
Comments and Suggestions for Authors
Hsien-Jen Chiu et al evaluates the cross-protection of a commercial lineage 7 PRRSV MLV for the challenge of lineage 1 and lineage 3 PRRSV. However, there are some deficiencies in the experimental design, and some details of the material method are unknown.
1. How to evaluate the cross-protection effect of PRRS vaccine against different virus strains, and what are the key factors in the experimental design that may affect the reliability of the results? What challenges does the high variability and diversity of PRRS viruses pose to vaccine development and application, and what solutions or recommendations have been taken? It needs to be supplemented and discussed.
2. Line 89-92 Is the virus titer of IM 1 ml the same as that of ID 0.2 ml vaccine? Vaccine enrichment? What are the specific titers?
3. In study 2 (Laboratory recovery virulence test), bronchoalveolar lavage fluid was used for viral isolation and viral genome detection. However, Experimental data were missing included analysis of negative results, such as bronchoalveolar lavage fluid for virus isolation and viral genome detection.
4. Which detection method should be used to identify different linage PRRS in the sample detection after PRRSV challenge? Additional detailed methods are suggested.
5. The background of the pigs in study 3 was unknown, and the virus carrying situation (viremia) and antibody level of the pigs before immunization were not detailed.
6. Line 399-401 ID-vaccinated pigs developed earlier infection control than the IM-vaccinated ones. It is recommended to analyze the reasons for the differences in the discussion.
Author Response
1. How to evaluate the cross-protection effect of PRRS vaccine against different virus strains, and what are the key factors in the experimental design that may affect the reliability of the results? What challenges does the high variability and diversity of PRRS viruses pose to vaccine development and application, and what solutions or recommendations have been taken? It needs to be supplemented and discussed.
Our response: Thank you for your kind comments and suggestion. We have substantially rewritten the below paragraph to provide greater clarity, context and respond to your queries. (Line 349-375)
2. Line 89-92 Is the virus titer of IM 1 ml the same as that of ID 0.2 ml vaccine? Vaccine enrichment? What are the specific titers?
Our response: Thank you for your query. We prepared the vaccine according to the manufacturer’s instruction. Effectively there is no difference in virus titer between the ID and IM dose and the only difference is the volume of injection. It is necessary to have a lower volume of injection in order to ensure the vaccine remains in the intradermal space. We have added additional information to clarify this point. (Line 87-93)
3. In study 2 (Laboratory recovery virulence test), bronchoalveolar lavage fluid was used for viral isolation and viral genome detection. However, Experimental data were missing included analysis of negative results, such as bronchoalveolar lavage fluid for virus isolation and viral genome detection.
Our response: We thank the Reviewer # 1 for pointing out this issue. We have modified this as suggested. (Line 277-278)
4. Which detection method should be used to identify different linage PRRS in the sample detection after PRRSV challenge? Additional detailed methods are suggested.
Our response: Thanks for the reviewer # 1’s precise concerns. Identify the different PRRSV lineages were based on the complete open reading frame 5 sequences (Yim-Im et al., Microbiol Spectr 2023, 11, 2916-2923). (Line 156-157)
5. The background of the pigs in study 3 was unknown, and the virus carrying situation (viremia) and antibody level of the pigs before immunization were not detailed.
Our response: We thank the reviewer # 1 for the insightful point. We have added a new Figure 11 for the interpretation of PRRSV Ab in the field. The PRRSV load in the field was showed in Figure 10. (Line 311-324, Figure 10; Line 329-334, Figure 11)
6. Line 399-401 ID-vaccinated pigs developed earlier infection control than the IM-vaccinated ones. It is recommended to analyze the reasons for the differences in the discussion.
Our response: Thank you for your suggestion. We have added additional material to clarify some possible reasons for this and supported it with the findings of other authors who have performed similar work with ID and IM vaccination routes. (Line 405-418)
Reviewer 2 Report
Comments and Suggestions for Authors
This manuscript addresses a critical issue in pig health, PRRSV control, with a focus on cross-protection of vaccines, which is of high relevance for regions with diverse PRRSV lineages. The authors made a robust experimental design, including laboratory and field trials, to evaluate vaccine efficacy under different conditions, which provides valuable insights into the safety of the vaccine, including its stability and lack of reversion to virulence. Overall, the manuscript is well-structured and clearly written, making the data easy to interpret. The findings contribute valuable knowledge to PRRSV management.
Minor comments:
1. Figure 8: At 12 weeks of age, are the differences in body weight between the groups significant? Please verify. I looked at the error bars and means for each group and didn't feel the differences were significant.
2. Line 370-377: The authors evaluate intradermal (ID) and intramuscular (IM) vaccination routes, but the differences in immune responses are not linked to long-term performance outcomes, leaving questions about the practical advantages of one route over the other. I suggest the authors can provide more insights into why ID vaccination induced higher antibody levels and lower tissue viral loads but did not translate to measurable performance advantages.
3. Line 378-385: The authors mention the inability to distinguish between vaccine and challenge strains using RT-qPCR, but it does not sufficiently explore the potential impact of this limitation on the interpretation of results. I suggest that the authors can address limitations in methodology more thoroughly.
Author Response
1. Figure 8: At 12 weeks of age, are the differences in body weight between the groups significant? Please verify. I looked at the error bars and means for each group and didn't feel the differences were significant.
Our response: We thank the reviewer # 2 for the insightful point. Comparison of the liveweight of pigs in each group at 2, 4, and 12 weeks of age were compared through two-way ANOVA. (Line 171, 177)
2. Line 370-377: The authors evaluate intradermal (ID) and intramuscular (IM) vaccination routes, but the differences in immune responses are not linked to long-term performance outcomes, leaving questions about the practical advantages of one route over the other. I suggest the authors can provide more insights into why ID vaccination induced higher antibody levels and lower tissue viral loads but did not translate to measurable performance advantages.
Our response: Thank you for your kind comments. This is a difficult question to answer but we have tried our best to provide a reply. We do believe that there are production benefits to be found with ID vaccination compared to IM vaccination. (Line 419-432)
3. Line 378-385: The authors mention the inability to distinguish between vaccine and challenge strains using RT-qPCR, but it does not sufficiently explore the potential impact of this limitation on the interpretation of results. I suggest that the authors can address limitations in methodology more thoroughly.
Our response: Thanks for the reviewer # 2’s precise concerns. It is indeed a good suggestion and we have rewritten text in this paragraph to address your concerns. (Line 433-444)
Reviewer 3 Report
Comments and Suggestions for Authors
The manuscript investigates the cross-protection provided by a lineage 7 PRRSV MLV against a Lineage 1 isolate under lab conditions and a Lineage 3 challenge under field conditions. Overall, the work is well-designed and executed. After carefully reviewing this work, I have the following observations:
1. I recommend proofreading the text for typos and grammatical errors.
2. The title is too long. Consider shortening it.
3. Material and methods: Include a schematic diagram summarizing the animal study design and treatments.
4. Material and methods: Consider dividing this section into subsections, such as animal study (1, 2, and 3), treatment, measurements, and statistical analysis.
5. Figures 1-9: remove the figure titles. The information is already mentioned in the figure legend.
Author Response
1. I recommend proofreading the text for typos and grammatical errors.
Our response: Thank you for your suggestion. We have sent the document for additional English proof reading and corrections. Small edits have been made for ease of reading and clarity (highlighted in red).
2. The title is too long. Consider shortening it.
Our response: We thank the reviewer # 3 for the insightful point. We have amended the title to “Lineage 7 Porcine Reproductive and Respiratory Syndrome Vaccine demonstrates cross protection against Lineage 1 and Lineage 3 strains” (Line 2-4)
3. Material and methods: Include a schematic diagram summarizing the animal study design and treatments.
Our response: Thank you for your kind comments. We have modified this as suggested. (Figure 1)
4. Material and methods: Consider dividing this section into subsections, such as animal study (1, 2, and 3), treatment, measurements, and statistical analysis.
Our response: We have modified this as suggested. (Line 95, 137, 154 and 176)
5. Figures 1-9: remove the figure titles. The information is already mentioned in the figure legend.
Our response: We have modified this as suggested.
Reviewer 4 Report
Comments and Suggestions for Authors
Dear all,
I was happy to read and review the manuscript provided.
The manuscript addresses an important issue, the Porcine Reproductive and Respiratory Syndrome Virus (PRRSV). This is important in swine industry and they use modified-live vaccines (MLV) for control. they showed that PrimePac® PRRS is safe to use, successfully providing cross-protection against contemporary lineage 1 and lineage 3 PRRSV strains from Taiwan.
The authors investigated the cross-protection provided by a lineage PRRSV MLV against a Lineage 1 isolate under laboratory conditions, and a Lineage 3 challenge under field conditions, in Taiwan.
There were three studies:
In the first study, thirty PRRS antibody-negative conventional piglets were vaccinated via the intramuscular (IM) or the intradermal (ID) route, the control group receiving a placebo. Then 4 weeks the animals were challenged with a Taiwanese lineage 1 strain.
In the second study the standard protocol for detection of reversion to virulence was applied to the vaccine strain. There were used sixteen specific pathogen free piglets.
In the third study, on an infected pig farm in Taiwan (lineage 3 strain), three hundred piglets were randomly selected and divided into 3 groups, each injected with either the PrimePac® PRRS vaccine via the IM or the ID route, or a placebo.
All the work showed that in the first study, both vaccinated groups demonstrated reduced viraemia, as compared to the control group. The second study demonstrated that the MLV strain is stable. In the third study, piglet mortality, average daily weight gain and pig stunting rate were significantly improved in the vaccinated groups compared to the control group.
I found that the introduction and discussion section were well presented and with sufficient references.
I believe the methods and the study design is well conducted.
Also results are fine, tables and figures are designed aquarately.
Conclusions support the results.
I recommend publication, since it is important to vaccinate!
Author Response
1. I was happy to read and review the manuscript provided. The manuscript addresses an important issue, the Porcine Reproductive and Respiratory Syndrome Virus (PRRSV). This is important in swine industry and they use modified-live vaccines (MLV) for control. they showed that PrimePac® PRRS is safe to use, successfully providing cross-protection against contemporary lineage 1 and lineage 3 PRRSV strains from Taiwan. The authors investigated the cross-protection provided by a lineage PRRSV MLV against a Lineage 1 isolate under laboratory conditions, and a Lineage 3 challenge under field conditions, in Taiwan.
Our response: We thank the reviewer # 4 for your encouraging words and support of our publication. We hope that this helps veterinarians have confidence to use PRRS MLV vaccines to control PRRS infections in Asia. We sincerely thank you for reviewing our publication and will keep working hard in this area.
Round 2
Reviewer 1 Report
Comments and Suggestions for Authors
Although the manuscript has made a lot of additions, it still needs to be modified and perfected.
1. Line 87-93 vaccine injection volume is different, vaccine virus titer should also be different, vaccine production batch number should be different.
2. Line 276-280 Since the virus was not detected in bronchoalveolar lavage fluid, why not isolate the virus or virus genome from lung tissue, lymph nodes, or blood for testing?
3. Figure 10. Different linage PRRS were not identified in the sample test after PRRSV challenge, and the changes of different linage PRRS titers in viremia could not be understood. Since it is clear that PRRS-positive pig farm is PRRSV lineage 3 isolate and PRRSV Lineage 7 vaccine is used, Some literatures have reported the differential real time PCR method of PRRSV lineage 3 and Lineage 7, which can evaluate the titer changes of PRRS Lineage 7 vaccine strain and PRRSV lineage 3 wild strain in viremia.
Tao C, Zhu X, Huang Y, Yuan W, Wang Z, Zhu H, Jia H. Development of a Multiplex RT-qPCR Method for the Identification and Lineage Typing of Porcine Reproductive and Respiratory Syndrome Virus. Int J Mol Sci. 2024 Dec 8; 25(23):13203. doi: 10.3390/ijms252313203. PMID: 39684913; PMCID: PMC11642648.
4. Figure 8. While data on pig deaths are provided, cases and mortality in each group of pigs should be provided
5. Figure 11. There was a high level of maternal PRRSV antibody in the piglet group before PRRSV vaccination. How about one degree? Is it suitable for vaccination? High levels of maternal antibodies may inhibit the replication of PPRS vaccine and affect the efficacy of the vaccine.
Author Response
Response to Reviewer # 1
- Line 87-93 vaccine injection volume is different, vaccine virus titer should also be different, vaccine production batch number should be different.
Our response: Thank you for your question. It is a fair question and to clarify, the vaccine is available as a single freeze dried cake for use in either IM or ID injection. Prior to use in the field, the manufacturer recommends to reconstitute with Diluvac Forte, an adjuvant either to 0.2mL per dose for ID or 1mL for IM. The vaccine virus titre is guaranteed by the manufacturer to be at least 4.0 log10 TCID50 per dose in either the IM or ID dosing. The vaccine production batch number has been added and it is same for both products. The only difference is the amount of adjuvant used to reconstitute the freeze dried cake. (Line 87-94)
- Line 276-280 Since the virus was not detected in bronchoalveolar lavage fluid, why not isolate the virus or virus genome from lung tissue, lymph nodes, or blood for testing?
Our response: Thank you for your comments. We have added additional wording to clarify our findings. Blood collected and tested via qPCR from the first passage was positive at CQ level 38 in one animal, with the rest negative. In the second passage, blood samples were all qPCR negative. Unfortunately, in this case CQ values were low, not enough to allow us to perform genome sequencing of the recovered virus. We acknowledge that because we could not recover the genome of the virus, we cannot assert that no genetic changes happened in the virus after passage and have modified the manuscript accordingly. We maintained the statement that there is minimal shedding of this vaccine strain when compared to field strains. (Line 380-382)
- Figure 10. Different linage PRRS were not identified in the sample test after PRRSV challenge, and the changes of different linage PRRS titers in viremia could not be understood. Since it is clear that PRRS-positive pig farm is PRRSV lineage 3 isolate and PRRSV Lineage 7 vaccine is used, Some literatures have reported the differential real time PCR method of PRRSV lineage 3 and Lineage 7, which can evaluate the titer changes of PRRS Lineage 7 vaccine strain and PRRSV lineage 3 wild strain in viremia.
Tao C, Zhu X, Huang Y, Yuan W, Wang Z, Zhu H, Jia H. Development of a Multiplex RT-qPCR Method for the Identification and Lineage Typing of Porcine Reproductive and Respiratory Syndrome Virus. Int J Mol Sci. 2024 Dec 8; 25(23):13203. doi: 10.3390/ijms252313203. PMID: 39684913; PMCID: PMC11642648.
Our response: Thank you for your suggestion. As we found, there was no statistically significant difference in viraemia among the three groups at any stage (Figure 10). We also demonstrated that despite this, only vaccinated groups in our field study had improved ADG and reduced mortality and stunting rate. However, this method you have shared could certainly have helped us to understand the background of the viremia in the different groups and is a valuable suggestion to improve our future studies. Unfortunately, we are not able to optimize this RT-qPCR in time for our revision deadline. We have accordingly cited this reference and listed our limitations in this study. (Line 450-467)
- Figure 8. While data on pig deaths are provided, cases and mortality in each group of pigs should be provided
Our response: Thank you for your comments. We have supplemented the information with additional data as suggested. We have updated Figure 8 accordingly too (Line 287-291, Figure 8)
- Figure 11. There was a high level of maternal PRRSV antibody in the piglet group before PRRSV vaccination. How about one degree? Is it suitable for vaccination? High levels of maternal antibodies may inhibit the replication of PPRS vaccine and affect the efficacy of the vaccine.
Our response: Thank you for your comments. We have added additional wording to respond to your comments. We believe that the maternal PRRSV antibody levels in all groups at the 2w mark was equally high. Despite the medium to high level, we demonstrated that we were able to improve clinical parameters such as ADG and mortality, demonstrating that vaccination is still a valuable tool in the face of medium to high levels of maternal PRRSV. (Line 408-421)
